# Diagnostic Accuracy of Ultrasound Guided Percutaneous Pleural Needle Biopsy for Malignant Pleural Mesothelioma

**DOI:** 10.3390/jcm13092600

**Published:** 2024-04-29

**Authors:** Carlo Iadevaia, Vito D’Agnano, Raffaella Pagliaro, Felice Nappi, Raffaella Lucci, Simona Massa, Andrea Bianco, Fabio Perrotta

**Affiliations:** 1U.O.C. Clinica Pneumologica L.Vanvitelli, Monaldi Hospital, A.O. dei Colli, 80131 Naples, Italy; carlo.iadevaia@aodeicolli.it (C.I.); vito.dagnano@studenti.unicampania.it (V.D.); raffaella.pagliaro@studenti.unicampania.it (R.P.); andrea.bianco@unicampania.it (A.B.); 2Department of Translational Medical Sciences, University of Campania L. Vanvitelli, 80131 Naples, Italy; 3Department of Respiratory Medicine, Boscotrecase COVID Hospital, 80042 Boscotrecase, Italy; felice.nappia@aslna3sud.it; 4Unit of Pathology, Monaldi Hospital, A.O. dei Colli, 80131 Naples, Italy; raffaella.lucci@aodeicolli.it (R.L.); simona.massa@aodeicolli.it (S.M.)

**Keywords:** malignant pleural mesothelioma, ultrasound, biopsy, diagnosis

## Abstract

**Background/Objectives**: Ultrasound (US) has been progressively spreading as the most useful technique for guiding biopsies and fine-needle aspirations that are performed percutaneously. Malignant pleural mesothelioma (MPM) represents the most common malignant pleural tumour. Thoracoscopy represents the gold standard for diagnosis, although conditions hampering such diagnostic approach often coexist. The Objective was to determine whether ultrasound-guided percutaneous needle biopsy (US-PPNB) has a high diagnostic accuracy and represents a safe option for diagnosis of MPM. **Methods:** US-PPNB of pleural lesions suspected for MPM in patients admitted from January 2021 to June 2023 have been retrospectively analyzed. An 18-gauge semi-automatic spring-loaded biopsy system (Medax Velox 2^®^) was used by experienced pneumologists. The obtained specimens were histologically evaluated and defined as adequate or non-adequate for diagnosis according to whether the material was considered appropriate or not for immunohistochemistry (IHC) analysis. The primary objective of the study was the diagnostic yield for a tissue diagnosis. **Results:** US-PPNB was diagnostic of MPM in 15 out of 18 patients (sensitivity: 83.39%; specificity: 100%; PPV: 100%). Three patients with non-adequate US-PPNB underwent thoracoscopy for diagnosis. We found significant differences in terms of mean pleural lesion thickness between patients with adequate and not-adequate biopsy (15.4 mm (SD: 9.19 mm) and 3.77 mm (SD: 0.60 mm), *p* < 0.0010. In addition, a significant positive correlation has been observed between diagnostic accuracy and FDG-PET avidity value. **Conclusions:** US-PPNB performed by a pneumologist represents a valid procedure with a high diagnostic yield and accuracy for the diagnosis of MPM, and may be considered as an alternative option in patients who are not suitable for thoracoscopy.

## 1. Introduction

Malignant pleural mesothelioma (MPM) represents a tumour of malignant mesothelial cells and the most frequent pleural neoplasia with three main pathological subtypes (epithelioid, sarcomatoid or biphasic) [1,2]. Unilateral pleural effusion (PE) is common, whilst presenting symptoms including chest pain, dyspnea, weight loss, are vague and not specific. Laboratory blood test, chest X-ray, and chest and abdomen computed tomography represent key steps in the standard work-up of patients with suspected mesothelioma. In addition, 18-F-fluorodeoxyglucose positron emission tomography/computed tomography (18-FDG-PET-CT) scan is considered a diagnostic aid in distinguishing malignant from benign lesions and detecting occult metastatic disease. Thoracoscopy remains the recommended choice to obtain adequate tissue sampling for pathological diagnosis [2]. However, the general performance status, comorbidities and poor ventilatory reserve may limit the utility of more invasive diagnostic procedures in clinical practice [3]. Despite the advances in thoracic ultrasound (US) for pleural cavity examination and for guiding biopsy procedures, small-cohort studies have investigated the role of imaging guided biopsies of pleural lesions with either CT scan or US [4,5,6]. In this scenario, as a non-ionizing approach and easily performed by pneumologists, US-guided biopsies may represent the procedure of choice in patients with suspected MPM not eligible for thoracoscopy.

The aim of the present study is therefore to assess the diagnostic yield and the potential complications of ultrasound–percutaneous pleural needle biopsy (US-PPNB) in a cohort of consecutive patients with suspected MPM and to evaluate the relevance of 18-FDG-PET-CT in influencing the diagnostic yield of US-PPNB.

## 2. Methods

### 2.1. Patients

We therefore present a retrospective single-center study including consecutive patients admitted from January 2021 to June 2023 at “Pneumologic Clinic L. Vanvitelli”, Monaldi Hospital, A.O. dei Colli, Naples, Italy. Patients with pleural thickening and a conclusive diagnosis of MPM have been included. Pleural thickening was defined as a CT finding of at least 3 mm, as previously described [7]. Demographic, clinical and radiological data from contrast-enhanced total body CT and 18-FDG-PET-CT scans were systematically recorded. All patients underwent two-dimensional (B-mode) thoracic lung ultrasound. Two probes were used: a curvilinear, convex, low-frequency (3–7 MHz) probe to evaluate pleural lesion depth, and a linear, high-frequency (7–15 MHz) probe for a more detailed examination of the pleural line. Anterior, lateral and posterior lung zones of each hemithorax have been explored according to current guidelines [8]. The transducer has been tilted and moved along the intercostal spaces. The thickness of the lesions has been assessed. Pleural lesions were considered thin with a thickness less than 4.15 mm. Those with a thickness equal to or more than 4.15 mm were defined as thick, according to Zhang et al. [9]. Colour Doppler function has been employed to confirm and exclude vascular structures. Exclusion criteria for biopsy are summarized in Table 1. Pain has been evaluated via the Numeric Rating Scale (NRS). 

### 2.2. US-PPNB: Description of Procedure

Procedure was carried out under moderate sedation with intravenous midazolam (1–3 mg) and local anaesthesia using injectable 2% lidocaine [10,11], in either the supine or the prone position depending on the site of the target lesion. An expert pulmonologist performed all the procedures. To ensure sterilization, sterile ultrasound probe covers were used. Skin disinfection was achieved using povidone–iodine (Betadine^®^) (10% PVP-I). Biopsies were performed using an 18-gauge semi-automatic spring-loaded biopsy system (Medax Velox 2^®^, Poggio Rusco, Italy) under either low-frequency (3–7 MHz) or high-frequency (7–15 MHz) ultrasound probe guidance according to both patient habitus and lesion features (Figure 1). A number of passes ranging from one to three were obtained from each patient. Tissue specimens obtained were expelled on a slide and immediately after laid in embedding cassettes for biopsy, fixed in 10% formalin and sent for histopathological examination. On-site May–Grünwald–Giemsa staining and examination of the obtained specimen was performed by a second trained respiratory physician, leading to an immediate and preliminary evaluation of the specimen. After the procedure, sonography of the chest using a linear probe was performed to exclude procedure-related complications. Chest X-ray and a complete blood count test after 3 h were obtained. The duration of procedure (DOP) was defined as the interval time between the initiation of sedation and the last post-procedure ultrasound. Informed consent was obtained from each patient prior to undergoing pleural biopsy. Ethical approval was not required considering the observational nature of the study. The study was conducted in accordance with the Declaration of Helsinki.

### 2.3. Assessment of Samples

Pathological findings were dichotomically defined as adequate or non-adequate for diagnosis according to whether material was considered appropriate or not for immunohistochemistry (IHC) analysis. Two different pathologists analyzed all the specimens obtained according to current WHO guidelines [2], including major subtype, architectural pattern, grading of epithelioid subtypes as well as stromal and cytological features. Depending on the morphology, we used immunohistochemical panels of positive and negative markers for mesothelial differentiation and for lesions considered in the differential diagnosis. An initial workup contained 2 mesothelial markers—WT1 (6F-H2) and anti-Calretinin (SP65) antibody—and 2 markers for the other tumor under consideration based on morphology (adenocarcinoma, squamous cell carcinoma), such as CEA (CEA31), Ep-CAM/Epithelial Specific Antigen (Ber-EP4) and AntibodyFactor-1 (8G7G3/1). If the results were concordant, the diagnosis was considered established. If they were discordant, a second stage, expanding the panel of antibodies, was performed. Additional antibodies were selected according to the differential diagnosis. Patients with not adequate US-PPNB samples subsequently underwent thoracoscopy [12]. 

### 2.4. Statistical Analysis

The primary aim of the study was to evaluate the diagnostic performance of US-PPNB in the diagnosis of MPM. The secondary endpoints were to define factors associated with adequate sampling. Proportions were compared using the chi-squared test or Spearman’s test as appropriate. Continuous variables were expressed as appropriate—median and interquartile range or mean and standard deviation—according to the distribution. Categorical data were expressed as number and percentage. The univariate analysis was performed using parametric (Student’s t for independent samples) and non-parametric (U-Mann–Whitney) statistics for continuous variables. Sensitivity, specificity and positive predictive value (PPV) were calculated as previously described [13]. An AUC to evaluate the effect of the 18-FDG-PET-CT avidity in influencing the diagnostic yield was provided. A *p*-value of <0.05 was considered as statistically significant. All analyses were performed using statistical software STATA v16 (StataCorp. 2019. StataCorp LLC, College Station, TX, USA).

## 3. Results

Between January 2021 and June 2023, 18 consecutive patients (11 (61%) males; 7 (39%) females) underwent ultrasound-guided pleural biopsy for suspected malignant pleural mesothelioma (Figure 2). The baseline characteristics and comorbidities of patients are summarized in Table 2. The mean age at the time of procedure was 71.2 (SD: ±6.61). In the study cohort, 11 patients (61%) reported asbestos exposure. Eleven patients (61%) had PE. All patients had an abnormal avidity of uptake at FDG-PET scan with a mean standardized uptake value (SUV) max of 10 (SD: ±1.58; range: 8–14). Median pleural lesion thickness was 13.5 mm (SD: ± 9.45 mm). Fourteen patients (78%) had thick lesions. Ultrasound contrast was not employed in any procedure. Mean Performance Status assessed by Eastern Cooperative Oncology Group (ECOG-PS) was 1.55. Systemic arterial hypertension (88%), COPD (38%) and chronic respiratory failure (38%) were the most frequent comorbidities. Data on the US-guided biopsies are shown in Table 3. Pleural biopsy was diagnostic of MPM in 15 patients (sensitivity: 83.39%; specificity: 100%; PPV: 100%). Pathological findings were consistent with epithelioid MPM in 12 patients (66%) (Figure 3a); 2 (11%) patients had biphasic MPM, and 1 had sarcomatoid (5%) (Figure 3a–f). In three patients (16%), the samples were not adequate for pathological assessment. These patients underwent thoracoscopy with a final diagnosis of MPM; in particular, two patients had sarcomatoid (11%), whilst one had epithelioid (5%). The mean pleural lesion thickness in patients with adequate and not-adequate biopsy was 15.4 mm (SD: ±9.19 mm) and 3.77 mm (SD: ±0.60 mm), respectively. Diagnostic accuracy was positively correlated with pleural lesion thickness (*p* < 0.001). A significant positive correlation has been observed between diagnostic accuracy and 18-FDG-PET-CT avidity value (SUVmax) (*p* = 0.012; Appendix A). Specifically, lower values of SUVmax were associated with worse diagnostic accuracy. Fourteen patients had a score less than 5 in the NRS, whilst only four of them had a score greater than 5. Mean DOP was 21 min (SD: ±4.67). Four patients experienced iatrogenic complications including pneumothorax (PNT). In all case, a small-bore chest tube Drentech UNICO™ was used and PNT was resolved within an average of 4 days (range: 2–7). No hemorrhagic complications have been recorded. 

## 4. Discussion

In patients with suspected mesothelioma, a prompt and adequate diagnosis is of paramount importance in order to minimize treatment delay. Thoracoscopy is considered as the gold standard biopsy technique for the diagnosis of MPM. However, dry circumferential pleural thickening without PE might cause the cavity to be inaccessible and thoracoscopy unfeasible [14]. Furthermore, general clinical conditions coupled with the burden of comorbidities in patients with thoracic malignancies may represent a major contraindication for such a diagnostic approach [15,16]. In addition, as the procedure requires an operating theatre space as well as the presence of an anesthetist—for those procedures which are performed under general anesthesia—we must consider the financial cost of thoracoscopy. In this scenario, bed-side pleural insonification allows a prompt assessment of suspected malignant pleural thickening, PE evaluation, as well as the identification of potential biopsy sites. Our study has demonstrated that the US-PPNB performed by a pneumologist is a safe and effective procedure in patients with suspected MPM, with a diagnostic yield of 83.39%. Our results are consistent with data from previous studies. In a cohort of 49 patients with clinically confirmed MPM, Zhang and coworkers demonstrated a diagnostic accuracy of 75.5% [9]. Variables impacting the diagnostic yield of US-guided pleural biopsies have been evaluated and should be considered in the decision-making process. In this regard, it has been reported that diagnostic accuracy was significantly higher when using a 16 G rather than an 18 G needle (*p* < 0.05) [9]. These data are similar to those reported by Haaga and collegues demonstrating a higher diagnostic yield with a larger needle [17]. However, Giuliani et al. did not find a significant difference between small (16 and 18 G) and 14 G needles in terms of diagnostic accuracy, in a total of 1118 ultrasound-guide core needle breast biopsy US-CNB cases [18]. Likewise, in a cohort of 70 patients with suspected MPM undergoing US-guided core needle biopsy, Heilo et al. found no significant differences in terms of diagnostic accuracy with regard to needle size as well as the number of punctures made in each procedure [19]. Pleural thickness may also play a crucial role. We have demonstrated a significant correlation between pleural thickness and diagnostic accuracy (*p* < 0.001). Specifically, patients with adequate biopsies had a mean pleural lesion of 15.4 mm (SD: ±9.19 mm) compared to patients with not-adequate biopsy whose mean pleural lesion thickness was of 3.77 mm (SD: ±0.60 mm). These findings appear consistent with data provided by Zhang et al. [9], which demonstrated that a pleural thickness greater than 4 mm ensures a significant diagnostic accuracy. In detail, the ROC curve showed a diagnostic accuracy reaching a value of 93.3% with a cut-off value of 4.15 mm (thick pleurae), compared to 47.4% in patients with thin pleurae (less than 4.15 mm) [9]. Another study investigating the factors influencing the diagnostic accuracy of US-guided pleural cutting needle biopsy reported that in patients with a pleural thickness less than 3 mm, the diagnostic yield was 61.0% versus 85.2% with a pleural thickness equal to or more than 3 mm (*p* = 0.001) [20].

The presence of PE is a clinical hallmark of pleural malignancy, reaching a prevalence of 94% in the affected side [21]. Able to detect effusion as small as 20 mL, US has a higher diagnostic accuracy when compared with chest X-rays (93% versus 47%) [22]. With the patient in a seated position, it also allows a more accurate quantification of fluid when compared to chest CT scans. Furthermore, it allows a real-time visualization of the needle during the procedure, such as during thoracentesis or biopsy, minimizing the risk of complications. However, a number of patients with MPM may not have PE at presentation. In this peculiar subset of patients, US-PPNB may represent the best approach to obtaining adequate samples. In a cohort of 20 patient with dry mesothelioma without PE, Stigt and colleagues demonstrated a diagnostic accuracy of 80% using a 14 G core needle biopsy mounted on an US probe, in line with our results [23]. It has also been demonstrated that mini thoracotomy and extrapleural biopsies may be considered as safe and accurate options for biopsy. In this scenario, TUS allows the identification of suspicious pleural thickness with a sensitivity and positive predictive value of 97% and 95%, respectively, as reported by Messina and coworkers [12]. Studies comparing efficacy and cost-effectiveness between mini thoracotomy and US-PPNB are sparse. In addition to the thickness of the pleural lesion, as previously mentioned, authors reported that ultrasonographic characteristics that may be suggestive of malignancy offer an irregular definition of knotty or planar pleural widening and echo-poor appearance [12]. Zhang and coworkers also found that pleural nodules as well as mass were significantly associated with a higher diagnostic yield in comparison with non-nodular pleural lesion (95.2% vs. 71.4%, *p* = 0.026) [20].

In patients with suspected MPM, FDG-PET-CT scans represent a valid diagnostic tool for the evaluation of primary and nodal extension, as well as occult distant metastasis [24,25]. Previous studies have demonstrated the utility of FDG-PET scans in discriminating pleural benign from malignant lesions [26], and current ESMO guidelines support its use especially for patients suited to macroscopic complete resection [27]. Moreover, growing evidence supports the role of FDG-PET/CT scanning in the assessment of response to treatment in patients with MPM, especially during treatment with immune checkpoint inhibitors (ICIs), which have represented a real game changer in the treatment of several malignancies [28,29,30]. According to our experience, all patients had an abnormal FDG-PET avidity value, and lower values were significantly associated with worse diagnostic accuracy. In this respect, FDG-PET avidity value should be considered in the choice of the most favorable site for biopsy, although false negative results at FDG-PET/CT scan have been reported in a cohort of 141 patients diagnosed with MPM [31].

Limitations of the study have been recognized. The two first issues are represented by the retrospective design of the study and the small number of enrolled patients. However, mesothelioma is a rare cancer, and regional prevalence deserves to be considered. In this respect, prospective multicenter studies are urged. Although useful for detecting vascular structures, the Doppler signal has not been correlated with diagnostic accuracy due to the poor quality of images. A final issue is that contrast-enhanced ultrasound (CEUS) has not been employed in our cohort. However, the utility of CEUS has been widely demonstrated in improving US-guided biopsies’ diagnostic accuracy, as well as reducing the complication rates [32,33], and its utilization should be encouraged whenever feasible.

In conclusion, we have demonstrated that 18-G US-PPNB, performed by a pneumologist, provides adequate samples for histological evaluation in patients with suspected MPM with a high diagnostic yield.

## Figures and Tables

**Figure 1 jcm-13-02600-f001:**
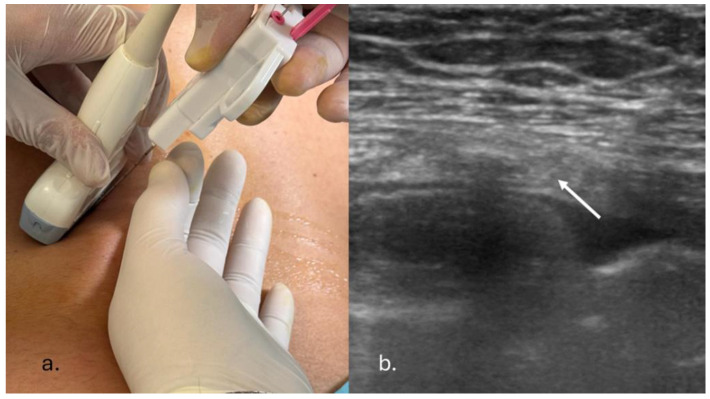
(**a**) Ultrasound-guided percutaneous pleural needle biopsy. A second physician holds the probe during the procedure. (**b**) Convex probe thoracic ultrasound-guided biopsy of a pleural lesion (needle is indicated with white arrow).

**Figure 2 jcm-13-02600-f002:**
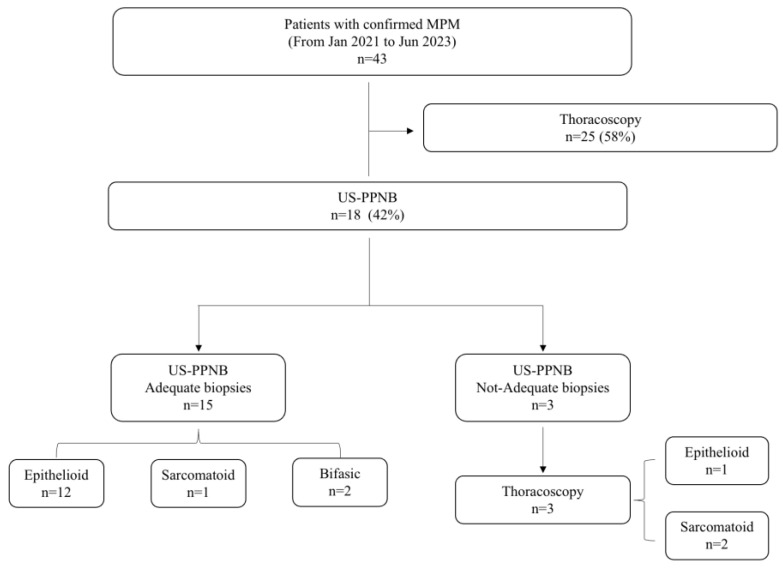
Flowchart of patients who underwent US-PPNB for suspected MPM. MPM: malignant pleural mesothelioma; US-PPNB: ultrasound–percutaneous pleural needle biopsy.

**Figure 3 jcm-13-02600-f003:**
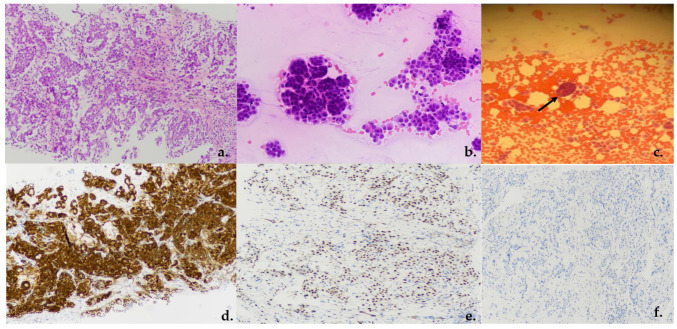
(**a**) US-PPNB, hematoxylin/eosin, epithelioid neoplasm with tubular papillary solid and trabecular pattern (magnification ×20); (**b**,**c**) cytology of rapid on-site evaluation—epithelioid cells arranged in sheets and morules ((**b**) magnification ×20 and (**c**) arrow shows magnification ×10). (**d**) IHC, tumoral cells showed positive staining for calretinin (magnification ×20). (**e**) IHC, tumoral cells showed positive nuclear staining for WT1 (magnification ×20). (**f**) IHC, tumoral cells showed negative staining for CEA (magnification ×20).

**Table 1 jcm-13-02600-t001:** Exclusion criteria for US-PPNB. AMS: Altered mental status. EF: ejection fraction. INR: International Normalized Ratio. P/F: PaO_2_/FiO_2_ ratio. PNT: pneumothorax.

Rib fractures
AMS
Uncontrolled cardiac arrythmia
PNT
Severe anemia
Chronic heart failure (FE < 40%)
Severe respiratory failure (P/F ratio < 250)
INR > 1.5

**Table 2 jcm-13-02600-t002:** Baseline characteristics and comorbidities. COPD: chronic obstructive pulmonary disease. ECOG: Eastern Cooperative Oncology Group. PS: performance status. SD: standard deviation. SUV: standardized uptake value.

Total number of patients	18
Males	11 (61%)
Median age (sd)	71.2 (±6.61)
Mean ECOG-PS	1.55
TNM	
I	3 (16%)
II–III	15 (84%)
Reported exposure to asbestos	11 (61%)
Mean pleural lesion thickness (sd)	13.5 mm (±9.45)
Mean SUVmax (sd)	10.7 (±1.58)
Pleural effusion	11 (61%)
Comorbidity	
Arterial hypertension	16 (88.8%)
Chronic renal failure	5 (27.8%)
COPD	7 (38.9%)
Chronic respiratory failure	7 (38.9%)
Pregress stroke	3 (16.7%)
Chronic ischemic disease	4 (22.2%)
Diabetes	3 (16.7%)
Prostatic hypertofic disease	6 (33.3%)

**Table 3 jcm-13-02600-t003:** Biopsy data on 18 patients with confirmed MPM. DOP: duration of procedure; NRS: Numerical Rating Scale; PNT: pneumothorax.

Mean number of samples (sd)	2 (±0.9)
Accuracy	
Adequate biopsies	15 (83%)
Not-adequate biopsies	3 (17%)
Histology	
Epithelioid	13 (72%)
Sarcomatoid	3 (17%)
Bifasic	2 (11%)
Adequate biopsies	
Epithelioid	12 (72%)
Sarcomatoid	1 (5%)
Bifasic	2 (11%)
Not-adequate biopsies	
Epithelioid	1 (5%)
Sarcomatoid	2 (11%)
Bifasic	0 (0%)
Pleural lesion thickness	
Adequate biopsies	15.4 mm (±9.19)
Not-adequate biopsies	3.77 mm (±0.60)
Pain assessment	
NRS < 5	14 (78%)
NRS > 5	4 (22%)
Complications	
PNT	4 (22%)
Hemorrhagic complication	0 (0%)
DOP	21 min (±4.67)

## Data Availability

Data are contained within the article.

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
