# Peer review of "Diagnostic Accuracy of Ultrasound Guided Percutaneous Pleural Needle Biopsy for Malignant Pleural Mesothelioma"

_jcm, 2024, doi:10.3390/jcm13092600_

Round 1

Reviewer 1 Report

Comments and Suggestions for Authors

The authors have done a research regarding the diagnostic accuracy of ultrasound guided percutaneous pleural needle biopsy for malignant pleural mesothelioma.

I would like to congratulate the authors for their study.

As the study is well written, I also have some suggestions regarding some parts of the manuscript.

ABSTRACT:

·        The abstract is a little bit too long. Try to make it a little easier for any potential reader. It has a lot of information that is beyond the scope of the abstract (for example in the introduction).

INTRODUCTION

·        The introduction part does not have any references. It means that everything that is written there is discovered by you? I suggest to set some references about what is written there.

·        The paper (beside the abstract) does not have an explicit objective. Please emphasize in this part the objective of your study.

METHODS

·        You are not specifying the type of the study.

·        Every sentence used in the Patients section is very short, and with very low information. Please try to restructure the materials and methods section, in order to have a flow for your ideas.

·        As you use names of medicine which was administered for the patient (example: midazolam) please write also the concentrations that were used.

·        Respiratory physician is not the best term to be used in the context.

·        Skin disinfection after procedure was All patients underwent a post procedure chest x-ray and a complete blood count test after 3 hours – this sentence does not have any sense. Please re-write.

·        WHO guidelines – please write which guidelines, the year in which it was released and the name for WHO (World Health Organization).

·        Please correct ICH – I think you wanted to write IHC.

·        Because you wrote about WHO guidelines (I think that you make a reference about the blue books) and about IHC, you should write in this part which antibodies did you include into your IHC assessment.

·        About IHC, you say in the beginning that you decided if the materials were adequate or not for IHC. After this, a few rows down, you say that all specimens underwent IHC. Please decide.

·        About “time to diagnosis” you refer twice both in patients part and both in assessment of the samples part.

RESULTS

·        The results part is difficult to follow. I suggest to present your results more graphical, not in the text way because you use a lot of numbers which are hard to follow.

·        There is no confidence level expressed.

·        Figure 2 is a screen shot of the figure. Please try to save it as an image and insert it in the final version (pay attention to the bullets in the left corner).

·        Regarding figure 3: Please attach a scale to each image to identify the magnification. Please make an adequate description for each image that is seen. There is no description about the cells. You only mention that the cells were positive for something. Which cells were positive? The tumoral ones? The inflamatorry ones? In IHC not everything that is positive means that there is a neoplasm. You may say that immunohistochemically, the tumoral cells were positive for XXX.

·        The only mention about the histopathological assessment is in this sentence: Pathological findings were consistent with epithelioid MPM in 12 patients (66%); 2 (11%) patients had biphasic MPM, 1 had sarcomatoid (5%) (Figure 3 a-f). But you do not present what is seen in those pictures and what is about. Please re-write and re-arrange the pictures.

DISCUSSION

·        In my opinion, the discussion part is written in a difficult way to follow.

CONCLUSION

·        The conclusion is not in concordance with your title. As in title you talk about accuracy in diagnostic of ultrasound guided percutaneous pleural needle biopsy, in conclusion you say that is a well-tolarated and rapid safe procedure. Please re-write.

BIBLIOGRAPHY

·        More references and more from the last 5 years should be considered.

Comments on the Quality of English Language

Many phrases are hard to follow and there are some typos.

Author Response

Dear Editor many thanks for the chance to revise the manuscript entitled “Diagnostic accuracy of ultrasound guided percutaneous pleural needle biopsy for malignant pleural mesothelioma” submitted on your Journal.

In this version, we made an extensive revision of the paper according the Referees indications. Main changes included:

  • A stronger definition of the aims and novelty of the research
  • An accurate description of methods and of the statistical plan
  • A comprehensive review of the introduction and the discussion

Point by point response

In particular:

The authors have done a research regarding the diagnostic accuracy of ultrasound guided percutaneous pleural needle biopsy for malignant pleural mesothelioma.

I would like to congratulate the authors for their study.

As the study is well written, I also have some suggestions regarding some parts of the manuscript.

ABSTRACT:

  • The abstract is a little bit too long. Try to make it a little easier for any potential reader. It has a lot of information that is beyond the scope of the abstract (for example in the introduction).

Response: We thanks the referee for this comment. Lines 30 – 50. Abstract has been extensively reviewed accordingly.

INTRODUCTION

  • The introduction part does not have any references. It means that everything that is written there is discovered by you? I suggest to set some references about what is written there.

Response: We thanks the referee for this comment. Line 64-81. References have been added and introduction reviewed accordingly.

  • The paper (beside the abstract) does not have an explicit objective. Please emphasize in this part the objective of your study.

Response: We thanks the referee for this very important comment. Line 77-79. Objectives of the study have been accordingly added in the introduction.

METHODS

  • You are not specifying the type of the study.

Response: We thanks the referee for this very important comment. Line 83-84. We fixed the issue.

  • Every sentence used in the Patients section is very short, and with very low information. Please try to restructure the materials and methods section, in order to have a flow for your ideas.

Response: We thanks the referee for this very important comment. Methods subparagraph has been extensively reviewed. We hope you may appreciate this novel version.

  • As you use names of medicine which was administered for the patient (example: midazolam) please write also the concentrations that were used.

Response: We thanks the referee for this very important comment . Line 102-103. We thank the reviewer for the comment. Midazolam dosage has been added. Literature references have been added accordingly.

  • Respiratory physician is not the best term to be used in the context.

Response: We thanks the referee for this comment. Line 104: Pulmonologist has been used accordingly. Sorry for the inaccuracy

  • Skin disinfection after procedure was All patients underwent a post procedure chest x-ray and a complete blood count test after 3 hours – this sentence does not have any sense. Please re-write.

Response: We thanks the referee for this comment. Line114-115. Errors have been fixed and sentence has been rephrased. Sorry for the inaccuracy

  • WHO guidelines – please write which guidelines, the year in which it was released and the name for WHO (World Health Organization).

Response: We thanks the referee for this comment. Line 131. Reference has been added. Sorry for the inaccuracy

  • Please correct ICH – I think you wanted to write IHC.

Response: We thanks the referee for this comment. Line132. Line has been fixed.

  • Because you wrote about WHO guidelines (I think that you make a reference about the blue books) and about IHC, you should write in this part which antibodies did you include into your IHC assessment.

Response: We thanks the referee for this valuable comment. Line 135-Line 142. We do agree about the lack of information. In the revised version, more data about tissue specimens IHC have been added. We hope you may appreciate these changes

  • About IHC, you say in the beginning that you decided if the materials were adequate or not for IHC. After this, a few rows down, you say that all specimens underwent IHC. Please decide.

Response: We thanks the referee for this valuable comment. Line 131-138. Lines needed clarification. Accordingly, specimen was considered adequate if sample was sufficient for performing IHC. Lines have been fixed accordingly.

  • About “time to diagnosis” you refer twice both in patients part and both in assessment of the samples part.

Response: We thanks the referee for this valuable comment. We removed “ time to diagnosis “ in the assessment of the samples as for other referee indications. It may depend on several institutional factors and would be not generesable therefore we decide to remove this item.

RESULTS

  • The results part is difficult to follow. I suggest to present your results more graphical, not in the text way because you use a lot of numbers which are hard to follow.

Response: We thanks the referee for this valuable comment. We have removed redundant results (which are include in the Tables) and we focused only on main study population characteristics to provide an overview to the readers of the subjects included in the analysis

  • There is no confidence level expressed.
  • Figure 2 is a screen shot of the figure. Please try to save it as an image and insert it in the final version (pay attention to the bullets in the left corner).

Response: We thanks the referee for this valuable comment. Figure 2 has been fixed.

  • Regarding figure 3: Please attach a scale to each image to identify the magnification. Please make an adequate description for each image that is seen. There is no description about the cells. You only mention that the cells were positive for something. Which cells were positive? The tumoral ones? The inflamatorry ones? In IHC not everything that is positive means that there is a neoplasm. You may say that immunohistochemically, the tumoral cells were positive for XXX.

Response: We thanks the referee for this valuable comment. Figure 3 has been fixed accordingly

  • The only mention about the histopathological assessment is in this sentence: Pathological findings were consistent with epithelioid MPM in 12 patients (66%); 2 (11%) patients had biphasic MPM, 1 had sarcomatoid (5%) (Figure 3 a-f). But you do not present what is seen in those pictures and what is about. Please re-write and re-arrange the pictures.

Response: We thanks the referee for this valuable comment. Pictures have been rearranged.

DISCUSSION

  • In my opinion, the discussion part is written in a difficult way to follow.

Response: We thanks the referee for this valuable comment. Discussion have been extensively reviewed highlighting the main findings of the research and comparing the results with other pre-existing data.

CONCLUSION

  • The conclusion is not in concordance with your title. As in title you talk about accuracy in diagnostic of ultrasound guided percutaneous pleural needle biopsy, in conclusion you say that is a well-tolarated and rapid safe procedure. Please re-write.

Response: We thanks the referee for this valuable comment.

BIBLIOGRAPHY

  • More references and more from the last 5 years should be considered.

Response: We thanks the referee for this valuable comment. We add references as suggested.

Reviewer 2 Report

Comments and Suggestions for Authors

Dear Authors,

I have read an article regarding the diagnostic accuracy of ultrasound-guided-percutaneous pleural needle biopsy for malignant pleural mesothelioma (MPM). I have several issues that need to be addressed:

1) First of all, why was ethical approval not needed in this study? The authors mentioned that since this is a retrospective study, ethics are not needed. However, this was not mentioned in the methodology section. Please clarify.

2) The methodology presented in the abstract is confusing and does not adhere to JCM formatting. Please correct this.

3) The introduction is not coherent without any rationale explanation of the gap in previous studies. Furthermore, there is no aim of the study as well as no citations in the introduction.

4) How did the authors determine "Metabolic active pleural thickening" as part of the inclusion criteria? Please explain.

5) Are there any pictures of how was the transducer placed? Please include them if there are some pictures of those procedures.

6) After Table 1, there is a sub-heading of the procedure. What kind of procedure are we talking about here?

7) The authors mentioned diagnostic accuracy in the title but did not measure sensitivity, specificity, positive predictive value, and negative predictive value. Instead, they measured time to diagnosis which has no relevance here. Please explain.

8) In this study, what is the index test being referred to?

9) Why did the patients who had inadequate biopsies undergo thoracoscopy? This was not mentioned in the methodology. Who conducted the thoracoscopies? Please explain in greater detail.

10) The authors did not mention any data on the side effects and yet concluded that this is "a safe procedure". Please explain.

Comments on the Quality of English Language

Extensive editing is needed. Some typo errors occur such as "Pregress" and "hypertrofic" in Table 1.

Author Response

Dear Editor many thanks for the chance to revise the manuscript entitled “Diagnostic accuracy of ultrasound guided percutaneous pleural needle biopsy for malignant pleural mesothelioma” submitted on your Journal.

In this version, we made an extensive revision of the paper according the Referees indications. Main changes included:

  • A stronger definition of the aims and novelty of the research
  • An accurate description of methods and of the statistical plan
  • A comprehensive review of the introduction and the discussion

Point by point response

In particular:

Dear Authors,

I have read an article regarding the diagnostic accuracy of ultrasound-guided-percutaneous pleural needle biopsy for malignant pleural mesothelioma (MPM). I have several issues that need to be addressed:

1) First of all, why was ethical approval not needed in this study? The authors mentioned that since this is a retrospective study, ethics are not needed. However, this was not mentioned in the methodology section. Please clarify.

Response: We thank the referee for this valuable comment. Lines 120-122. Lines have been added to address the issue. In the methods paragraph we added the Ethics issue. Data have been treated according Helsinki declaration and informed consent was routinely obtained. The Ethical committee approval was waived as data have been completely anonymized and the study was a retrospective / clinical chart data review.

2) The methodology presented in the abstract is confusing and does not adhere to JCM formatting. Please correct this.

Response: We thank the referee for the comment. Abstract has been reviewed and corrected according to JCM guidelines for authors.

3) The introduction is not coherent without any rationale explanation of the gap in previous studies. Furthermore, there is no aim of the study as well as no citations in the introduction.

Response: We thank the reviewer for this important comment. Line 64-81. Introduction has been extensively reviewed as suggested. In addition, citations have been added in this part.

4) How did the authors determine "Metabolic active pleural thickening" as part of the inclusion criteria? Please explain.

Response: Line 86. We thank the reviewer for this important comment. As written in previous version of the manuscript, line might be misleading for the readers. We changed and fixed the line accordingly. As reported in the figure 2, consecutive patients with suspected pleural thickening have been considered for inclusion. An FDG PET-CT scan was performed, and avidity reported. 

5) Are there any pictures of how was the transducer placed? Please include them if there are some pictures of those procedures.

Response: We thank the reviewer for the comment. A picture has been added (Figure 1a)

6) After Table 1, there is a sub-heading of the procedure. What kind of procedure are we talking about here?

Response. We thank the referee for this important comment. Line 103: Procedure changed in “US-PPNB: description of procedure”. Sorry for the inaccurancy.

7) The authors mentioned diagnostic accuracy in the title but did not measure sensitivity, specificity, positive predictive value, and negative predictive value. Instead, they measured time to diagnosis which has no relevance here. Please explain.

Response. Line 165. Time to diagnosis has been accordingly removed.  Instead, sensitivity, specitivity, and positive predicted value have been added as suggested.

8) In this study, what is the index test being referred to?

Response. We thank the referee for this relevant comment. Line 70. We highlighted the importance of thoracoscopy as recommended choice to obtain tissue sampling.

9) Why did the patients who had inadequate biopsies undergo thoracoscopy? This was not mentioned in the methodology. Who conducted the thoracoscopies? Please explain in greater detail.

Response. Line 144-145. We thank the reviewer for the comment. If US-PPNB was not adequate for biopsy, patients underwent a surgical thoracoscopy (if Pleural effusion was present) or surgical biopsy, if PE was absent. Ref has been added accordingly. 

10) The authors did not mention any data on the side effects and yet concluded that this is "a safe procedure". Please explain.

Response. We thank the reviewer for the comment. 178-180. Complications have been described separately as suggested. In addition, it has also been reported in table 3.

Reviewer 3 Report

Comments and Suggestions for Authors

The manuscript is generally well-written, but certain parts require clarification. In their article, Iadevaia et al. investigate using pleural ultrasound to diagnose pleural mesothelioma. The analysis seems to be well-planned, with clear discussion and short conclusions. The authors demonstrated that 18-G ultrasound-guided pleural biopsy performed by a pneumologist is a rapid, safe procedure and provides adequate samples for histological evaluation in patients with suspected MPM; pleural thickening and PET avidity may influence the diagnostic yield.

Due to the relatively small number of cases and the retrospective study design, this study can be evaluated more like a pilot study; however, they show very encouraging results with a surprisingly low complication rate, which will require further studies.

Major comments:

It is very important that diagnostic interventions with as few complications as possible are conducted in the older, fragile patient group.

Do you have data regarding the performance status of patients?

In the histological examination of samples of FDG avid pleural thickening, did you find any other primary tumor, i.e., breast cancer or lung cancer?

Author Response

Dear Editor many thanks for the chance to revise the manuscript entitled “Diagnostic accuracy of ultrasound guided percutaneous pleural needle biopsy for malignant pleural mesothelioma” submitted on your Journal.

In this version, we made an extensive revision of the paper according the Referees indications. Main changes included:

  • A stronger definition of the aims and novelty of the research
  • An accurate description of methods and of the statistical plan
  • A comprehensive review of the introduction and the discussion

Point by point response

In particular:

Reviewer 3

The manuscript is generally well-written, but certain parts require clarification. In their article, Iadevaia et al. investigate using pleural ultrasound to diagnose pleural mesothelioma. The analysis seems to be well-planned, with clear discussion and short conclusions. The authors demonstrated that 18-G ultrasound-guided pleural biopsy performed by a pneumologist is a rapid, safe procedure and provides adequate samples for histological evaluation in patients with suspected MPM; pleural thickening and PET avidity may influence the diagnostic yield.

Due to the relatively small number of cases and the retrospective study design, this study can be evaluated more like a pilot study; however, they show very encouraging results with a surprisingly low complication rate, which will require further studies.

Major comments:

It is very important that diagnostic interventions with as few complications as possible are conducted in the older, fragile patient group.

Do you have data regarding the performance status of patients?

Response: We thanks the referee for this valuable comment . We thank the reviewer for the comment. Data regarding performance status have been added.

In the histological examination of samples of FDG avid pleural thickening, did you find any other primary tumor, i.e., breast cancer or lung cancer?

Response: We thanks the referee for this valuable comment. We only included data about malignant mesothelioma in our cohort. In this research article we aim to provide data only about MPM and not other malignant pleural thickening. As in the previous methods section it was not clear (leading to potential misinterpretation) we have now better described the revised version of the manuscript.

Reviewer 4 Report

Comments and Suggestions for Authors

The work is interesting, however can be improved. Please find the suggestions:

1) The abstract can be improved and the emphasis on the objectives should be more

2) "Two probes have been used: curvilinear, convex, low frequency (3-7 MHz) probe to evaluate pleural lesion depth whilst linear, high frequency (7-15 MHz) probe for a more detailed examination of the pleural line" What is the reason to use a curvilinear probe as compared to a linear probe?

3) Figure 2 may be better represented.

4) According to table 2, are 18 numbers of patients enough to conclude? Please explain.

5) "Our study has demonstrated that the US-PPNB performed by a pneumologist is a safe and efficacy procedure in patients with suspected MPM, with a diagnostic yield of 83.39%." How did the authors arrive at the number 83.39?

6) Discussions generally don't carry references. Please try to cite the papers before.

7) The manuscript can be better arranged. The authors may try to arrange the manuscript better.

8) Since this work is on ultrasound, the authors may cite recent advancements: "A modular approach to neonatal whole-brain photoacoustic imaging" DOI: /10.1117/12.2546854

Author Response

Dear Editor many thanks for the chance to revise the manuscript entitled “Diagnostic accuracy of ultrasound guided percutaneous pleural needle biopsy for malignant pleural mesothelioma” submitted on your Journal.

In this version, we made an extensive revision of the paper according the Referees indications. Main changes included:

  • A stronger definition of the aims and novelty of the research
  • An accurate description of methods and of the statistical plan
  • A comprehensive review of the introduction and the discussion

Point by point response

In particular:

Reviewer 4

The work is interesting, however can be improved. Please find the suggestions:

  • The abstract can be improved and the emphasis on the objectives should be more.

Response: We thanks the referee for this very important comment. Abstract has been reviewed accordingly. In addition in Lines 77-79. Objectives of the study have been accordingly added in the introduction.

  • "Two probes have been used: curvilinear, convex, low frequency (3-7 MHz) probe to evaluate pleural lesion depth whilst linear, high frequency (7-15 MHz) probe for a more detailed examination of the pleural line" What is the reason to use a curvilinear probe as compared to a linear probe?

Response: We thanks the referee for this very important comment. The convex probe ( as reported in the figure 1) has been used to evaluate the depth of the lesions and to evaluate pleural effusion, when it was present. In the clinical practise patients habitus , chest circumference, muscle or adipose thoracic tissue influence the choice of the probe.

  • Figure 2 may be better represented.

Response: We thanks the referee for this very important comment. Figure 2 has been reviewed to be easily readable.

4) According to table 2, are 18 numbers of patients enough to conclude? Please explain.

Response: We thanks the referee for this valuable comment. We do agree that the small number of patients represents a limit of the study, as reported in lines 266-268. However, MPM remain a rare tumour and at this time poor studies have been published. We agree that our data are not exhaustive; however, we corroborated previous results and we hope to prompt further larger trials. Finally we also originally describe the FDG avidity as potential factor influencing the diagnostic yield. 

5) "Our study has demonstrated that the US-PPNB performed by a pneumologist is a safe and efficacy procedure in patients with suspected MPM, with a diagnostic yield of 83.39%." How did the authors arrive at the number 83.39?

Response: We thanks the referee for this valuable comment. Line 147-148. Data have been extended as suggested, adding sensitivity, specificity and positive predicted value including a reference for the formulas.

6) Discussions generally don't carry references. Please try to cite the papers before.

Response: We thanks the referee for this valuable comment. As suggested, we add appropriate references in the introduction section.

7) The manuscript can be better arranged. The authors may try to arrange the manuscript better.

Response: We thanks the referee for this valuable comment. We extensively reviewed the manuscript as suggested.

8) Since this work is on ultrasound, the authors may cite recent advancements: "A modular approach to neonatal whole-brain photoacoustic imaging" DOI: /10.1117/12.2546854

Response: We thanks the referee for this valuable comment. We thank the reviewer for the comment. The ref has been added.

Round 2

Reviewer 1 Report

Comments and Suggestions for Authors

Dear authors,

The improvements made had a significant impact on your manuscript. Congrats.

All my concerns were answered.

Comments on the Quality of English Language

-

Author Response

Many thanks for your help in improving the quality of the manuscript

Reviewer 2 Report

Comments and Suggestions for Authors

Dear Authors,

I have assessed the revision. I still have a few more questions

1) If this is a retrospective review, I believe that more mesothelioma patients do not undergo US-PPNB. Please also mention them in the CONSORT diagram.

2) Again, how was pleural thickening defined?

3) Why was negative predictive value not included in this manuscript?

Comments on the Quality of English Language

-

Author Response

Dear Referee, 

Many thanks for your revision of the manuscript submitted. 

I have assessed the revision. I still have a few more questions

1) If this is a retrospective review, I believe that more mesothelioma patients do not undergo US-PPNB. Please also mention them in the CONSORT diagram.

The response tracked with yellow in the manuscript file.

1) Response : Many thanks for this valuable comment. We have reviewed the MPM diagnosis at our institution. We have included the number of MPM underwent to thoracoscopy (without a previous US-PNNB) in the CONSORT diagram

2) Again, how was pleural thickening defined?

2) Response: many thanks for this valuable comment. We have added a reference about pleural thickening definition. While there is no universal standard consensus, a CT finding more than 3 mm was previously described (Stark, P. Imaging in Pulmonary Disease; Goldman, L., Schafer, Editors. Goldman's Cecil medicine. Amsterdam: Elsevier; 2012. Chapter 84; ISBN 978-1-4377-1604-7). Highlighted in yellow in Methods paragraph the definition with reference. We are very sorry as we did not answer previously to this comment. 

3) Why was negative predictive value not included in this manuscript?

3) Response: NPV was not present as we have any false positive (patients with wrong diagnosis of MPM) in our cohort. 

Reviewer 4 Report

Comments and Suggestions for Authors

Paper is good to go

Author Response

(The authors gave the same response as above.)

Round 3

Reviewer 2 Report

Comments and Suggestions for Authors

-

Comments on the Quality of English Language

-